# Stepwise wiring of the *Drosophila* olfactory map requires specific Plexin B levels

Jiefu Li[1], Ricardo Guajardo[1], Chuanyun Xu[1], Bing Wu[1], Hongjie Li[1], Tongchao Li[1], David J Luginbuhl[1], Xiaojun Xie[2], Liqun Luo[1]*

[1]Department of Biology, Howard Hughes Medical Institute, Stanford University, Stanford, United States; [2]The Solomon H. Snyder Department of Neuroscience, Howard Hughes Medical Institute, The Johns Hopkins University School of Medicine, Baltimore, United States

**Abstract** The precise assembly of a neural circuit involves many consecutive steps. The conflict between a limited number of wiring molecules and the complexity of the neural network impels each molecule to execute multiple functions at different steps. Here, we examined the cell-type specific distribution of endogenous levels of axon guidance receptor Plexin B (PlexB) in the developing antennal lobe, the first olfactory processing center in *Drosophila*. We found that different classes of olfactory receptor neurons (ORNs) express PlexB at different levels in two wiring steps – axonal trajectory choice and subsequent target selection. In line with its temporally distinct patterns, the proper levels of PlexB control both steps in succession. Genetic interactions further revealed that the effect of high-level PlexB is antagonized by its canonical partner Sema2b. Thus, PlexB plays a multifaceted role in instructing the assembly of the *Drosophila* olfactory circuit through temporally-regulated expression patterns and expression level-dependent effects.

DOI: https://doi.org/10.7554/eLife.39088.001

*For correspondence:
lluo@stanford.edu

**Competing interests:** The authors declare that no competing interests exist.

## Introduction

Precise neural circuit assembly involves multiple coordinated steps. Through the division and differentiation processes, each postmitotic neuron obtains a specific fate that ultimately controls its wiring specificity and functional output. Axons and dendrites of a neuron then extend through specific paths to their gross targeting areas. As terminal structures elaborate, interactions between prospective synaptic partners finalize the connectivity selection and initiate synaptogenesis. Finally, plasticity of established circuits serves as an extended developmental mechanism, allowing dynamic adaptation for diverse neural functions. Studies in the past few decades have identified a large collection of molecules and revealed many developmental principles underlying the steps described above (*Hong and Luo, 2014*; *Hübener and Bonhoeffer, 2014*; *Jan and Jan, 2010*; *Jukam and Desplan, 2010*; *Kolodkin and Tessier-Lavigne, 2011*; *Li et al., 2018*; *Sanes and Yamagata, 2009*; *Zipursky and Sanes, 2010*). Nevertheless, only a limited number of guidance molecules are available compared to the complexity and heterogeneity of the neural network established under their instructions. Although each molecule can play several roles and function repeatedly at multiple stages, it is not well understood how a single molecule achieves this functional versatility and connects the sequential steps of neural circuit assembly.

The antennal lobe, the first-order olfactory processing center of *Drosophila*, comprises about 50 distinct glomeruli with stereotyped positions and morphologies and provides an excellent system for dissecting the molecular and cellular mechanisms underlying circuit assembly. Each of the 50 classes of olfactory receptor neurons (ORNs) expresses a single or a unique set of olfactory receptors and

projects their axons accurately to the 50 corresponding glomeruli in the antennal lobe (*Benton et al., 2009*; *Couto et al., 2005*; *Fishilevich and Vosshall, 2005*; *Gao et al., 2000*; *Silbering et al., 2011*; *Vosshall and Stocker, 2007*; *Vosshall et al., 2000*). Most projection neurons (PNs) arborize their dendrites in a specific glomerulus and form synaptic connections exclusively with the ORNs innervating that glomerulus (*Jefferis et al., 2001*; *Stocker et al., 1990*). Thus, the *Drosophila* olfactory map features 50 anatomically distinct information processing channels that ensure the fidelity of olfactory perception through wiring specificity – in the form of precise one-to-one pairings of 50 classes of ORNs and PNs.

Plexins are evolutionarily conserved cell-surface receptors that serve as the principal signaling route for semaphorins in the nervous system. The semaphorin-plexin signaling axis regulates many aspects of neural development, including axon guidance, dendrite targeting, synapse formation, and circuit plasticity (*Alto and Terman, 2017*; *Kolodkin et al., 1993*; *1992*; *Koropouli and Kolodkin, 2014*; *Kruger et al., 2005*; *Luo et al., 1993*; *Meltzer et al., 2016*; *Orr et al., 2017*; *Pascoe et al., 2015*; *Pasterkamp, 2012*; *Wang et al., 2017*). In the developing *Drosophila* olfactory circuit, degenerating larval ORNs build up a gradient of secreted Sema2a and Sema2b: high in the ventromedial (VM) and low in the dorsolateral (DL) antennal lobe. Transmembrane Sema1a forms an opposite gradient and functions cell-autonomously as a receptor to instruct PN dendrite targeting along the VM-DL axis (*Komiyama et al., 2007*; *Sweeney et al., 2011*). Moreover, this Sema2a/2b gradient also determines ORN axon trajectory choice through their canonical receptor PlexB (*Joo et al., 2013*).

At about 18 hr after puparium formation (APF), ORN axons arrive at the ventrolateral corner of the antennal lobe and then bifurcate into the DL and VM trajectories (*Jefferis et al., 2004*). Within the next 6 hr, ORN axons form two stereotyped bundles circumnavigating the antennal lobe, with a mean DL-to-VM ratio of 0.73 (*Figure 1A,C*) (*Joo et al., 2013*). Genetic disruption of *Sema2b* or *PlexB* biases ORN axons toward the DL trajectory, which subsequently leads to incorrect glomerular targeting (*Joo et al., 2013*), highlighting the significance of this preceding step in the precise assembly of the olfactory map. However, a mechanistic view of PlexB's functions in different steps of olfactory circuit assembly is still lacking. By devising an endogenous and conditional tag of PlexB proteins, we now observe the temporally transitory distribution of PlexB proteins in ORN axons in developing antennal lobes. Consistent with its spatial distribution, the expression level of PlexB controls both early-stage ORN axon trajectory choice and subsequent glomerular selection. Furthermore, the effect of high-level PlexB is antagonized by Sema2b, revealing another new aspect of this multi-functional molecule. We thus uncover how a single guidance molecule, PlexB, connects multiple steps of neural wiring by stage-specific distribution and level-dependent effects.

## Results

### Conditional tagging reveals that ORN axons of the DL trajectory possess a higher level of PlexB than the VM axons

Previously, we observed that ORN axons are significantly shifted to the DL trajectory in *plexB* mutant brains, leading to a mean DL-to-VM ratio of 1.53 in contrast to 0.73 in *wild-type* brains (*Figure 1B,C*) (*Joo et al., 2013*). Supplying PlexB in ORNs largely restores the normal trajectory pattern in *plexB* mutant flies, indicating that PlexB functions in ORNs to control the DL versus VM trajectory choice (*Joo et al., 2013*). However, the expression pattern of PlexB remains unknown because none of the PlexB antibodies could detect the endogenous PlexB signal in developing pupal brains (*Joo et al., 2013*), setting an obstacle for further dissection of its mechanisms of action in axon trajectory choice.

To reveal the PlexB expression pattern, we used a transcriptional reporter, *PlexB-GAL4* (Xiaojun Xie and Alex Kolodkin, unpublished), in which an artificial exon containing a splicing acceptor, an in-frame *T2A-GAL4* cassette, and an *Hsp70* terminator (*Diao et al., 2015*), was inserted into a MiMIC locus (*Venken et al., 2011*) in the first coding intron of *PlexB* (*Figure 1D*). This *PlexB-GAL4* thus hijacks the endogenous *PlexB* splicing program to produce a truncated PlexB fragment encoded by the first exon, and a GAL4 released from the PlexB fragment by T2A-mediated cleavage. Consistently, *PlexB-GAL4/plexB⁻* flies were sick and barely survived to adult stage, like *plexB⁻/⁻* homozygotes. When crossed to a membrane-bound fluorescent reporter line (*UAS-mCD8-GFP*), *PlexB-GAL4* labeled many structures in developing pupal brains, including the antennal lobe (*Figure 1E*). At

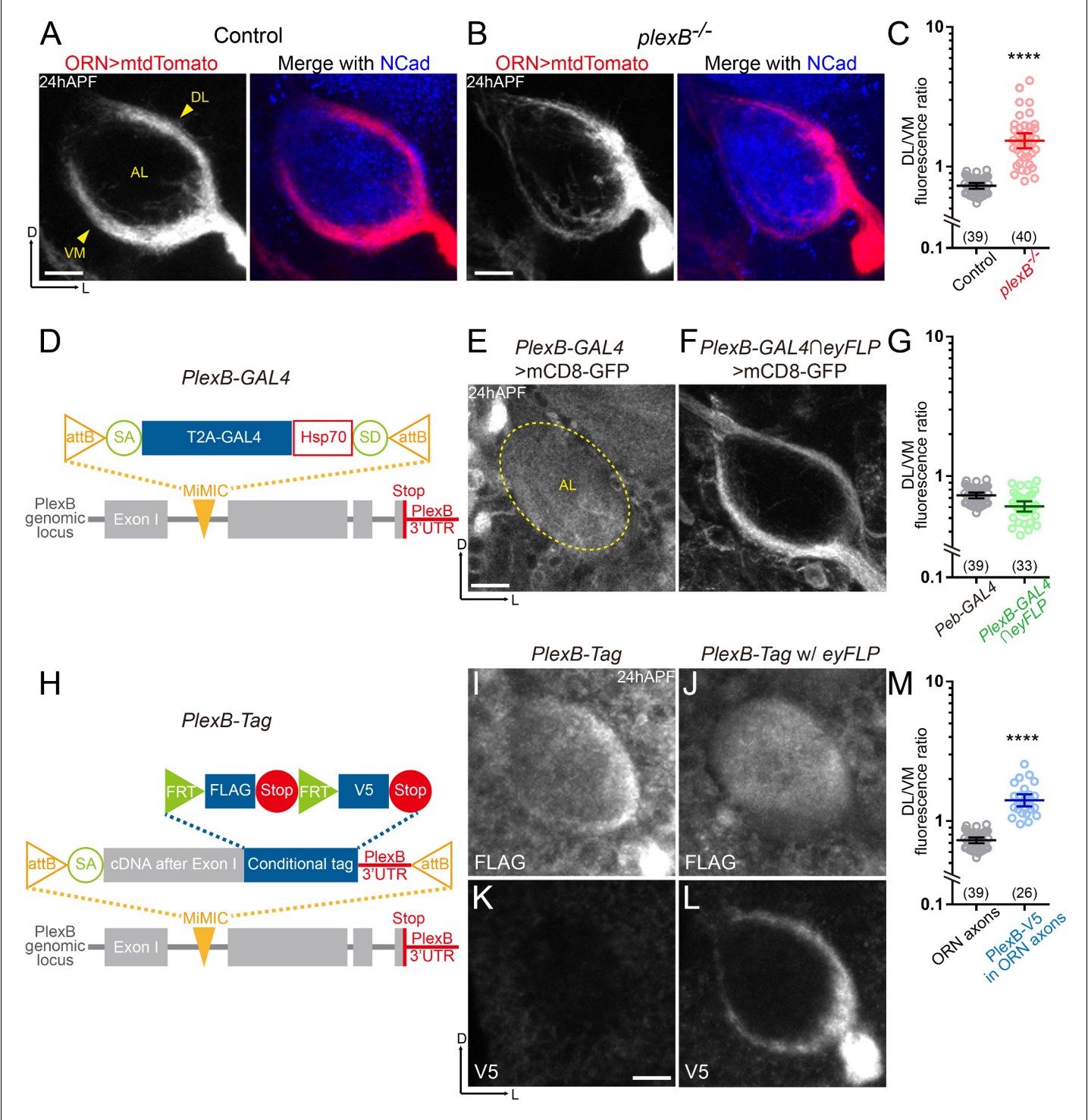

**Figure 1.** ORN axons of the DL trajectory express a higher level of PlexB proteins than the VM axons. (A) In *wild-type* pupal brains at 24 hr after puparium formation (24hAPF), ORN axons bifurcate and form the dorsolateral (DL) and ventromedial (VM) trajectories (arrowheads) circumscribing the antennal lobe (AL). ORN axons were labeled by the pan-ORN *Peb-GAL4* (***Sweeney et al., 2007***) driven mtdTomato expression. Antennal lobes were co-stained with a neuropil marker N-cadherin (NCad). (B) In *plexB* homozygous mutant, ORN axons preferentially choose the DL trajectory. (C) Fluorescence intensity ratios of ORN axon trajectories (DL/VM) in *wild-type* and *plexB$^{-/-}$* brains at 24hAPF. Geometric means: control, 0.73; *plexB$^{-/-}$*, 1.53. (D) Design of the *PlexB-GAL4*: a *T2A-GAL4* cassette (***Diao et al., 2015***) was inserted into a MiMIC locus in the first coding intron of *PlexB*. SA, splicing acceptor. Hsp70, terminator sequence of *Hsp70*. SD, splicing donor (not functional because mRNAs terminate at the *Hsp70* terminator). (E) *PlexB-GAL4* labels the antennal lobe and nearby brain structures at 24hAPF. The signal from ORNs is not detectable. Dotted circle, antennal lobe (AL).
*Figure 1 continued on next page*

*Figure 1 continued*

(**F**) Intersection of *PlexB-GAL4* and *eyFLP* (FLP in ORNs) labels both DL and VM trajectories. (**G**) DL/VM fluorescence intensity ratios of the pan-ORN GAL4 (*Peb-GAL4*) and the *PlexB-GAL4* intersected with *eyFLP*. Geometric means: *Peb-GAL4*, 0.73; *PlexB-GAL4∩eyFLP*, 0.61. (**H**) Design of the PlexB conditional tag: a cassette including a splicing acceptor (SA), the *PlexB* coding sequence after the first exon, *FRT-FLAG-Stop-FRT-V5-Stop*, and the *PlexB* 3'-UTR sequence, was inserted into a MiMIC locus in the first coding intron of *PlexB*. (**I, J**) FLAG staining of 24hAPF brains of *PlexB-Tag* alone (**I**) or *PlexB-Tag* with *eyFLP* (**J**). (**K, L**) V5 staining of 24hAPF brains of *PlexB-Tag* alone (**K**) or *PlexB-Tag* with *eyFLP* (**L**). (**M**) DL/VM fluorescence intensity ratios: ORN axon distributions were calculated based on the *Peb-GAL4 > mtdTomato* signal (as in *Figure 1A*); ORN-specific PlexB protein distributions were calculated based on the V5 staining of *PlexB-Tag* with *eyFLP* (as in *Figure 1L*). Geometric means: ORN axon distribution, 0.73; ORN-specific PlexB protein distribution, 1.41. Sample sizes are noted in parentheses. Significance between two groups was determined by an unpaired, two-tailed *t*-test. ****p<0.0001. Images are shown as maximum z-projections of confocal stacks. Scale bars, 10 µm. Axes, D (dorsal), L (lateral).
DOI: https://doi.org/10.7554/eLife.39088.002

The following figure supplement is available for figure 1:

**Figure supplement 1.** *PlexB-Tag* functions normally as the wild-type *PlexB*, and *PlexB-GAL4* labels vast majority of ORNs at 24hAPF.
DOI: https://doi.org/10.7554/eLife.39088.003

24hAPF, ORN axons circumscribe, but have not innervated, the antennal lobe, indicating that the signal inside of the antennal lobe was contributed by other types of cells such as PNs, whose dendrites innervate the antennal lobe ahead of ORN axons (*Jefferis et al., 2004*). To visualize the PlexB expression pattern specifically in ORNs, we employed an intersectional strategy in which the expression of mCD8-GFP was dual-gated by *PlexB-GAL4* and *eyFLP*, a FLP recombinase line only active in ORNs in the olfactory circuit. We observed that *PlexB-GAL4* labeled both DL and VM trajectories (*Figure 1F*). The DL-to-VM fluorescence ratio of *PlexB-GAL4* was similar to but slightly lower than that of pan-ORN *Peb-GAL4* (*Figure 1G*). Thus, *PlexB* is transcribed in both DL and VM ORNs. Indeed, by examining the third segments of antennae that house ORN somas, we found that *PlexB-GAL4* labeled > 90% of all ORNs (*Figure 1—figure supplement 1D*).

Discrepancies between transcription and protein levels due to post-transcriptional and post-translational regulations have been widely reported (*Carlyle et al., 2017*; *Liu et al., 2016*), necessitating an investigation of the endogenous protein distribution. Considering our failed attempts to visualize PlexB proteins using multiple custom antibodies, we reasoned that a knock-in protein tag of endogenous PlexB would likely yield a better outcome since monoclonal antibodies with high specificity are available for common tags such as FLAG and V5. As we observed in the *PlexB-GAL4* labeling, non-ORN cells also expressed PlexB, precluding our inspection of ORN-specific PlexB signal unless an intersectional strategy was taken. Inspired by the tagged Sema1a (*Pecot et al., 2013*), we devised a conditional tag of PlexB, hereafter *PlexB-Tag*, by inserting an artificial exon containing the PlexB coding sequence after the first exon and a conditional tag cassette (*FRT-FLAG-Stop-FRT-V5-Stop*) into the *PlexB* MiMIC locus (*Figure 1H*). To preserve any potential regulation by the 3'-UTR region, the original *PlexB* 3'-UTR was used instead of a generic terminator. This *PlexB-Tag* should produce full-length *wild-type* PlexB proteins tagged with FLAG or V5 epitopes at their C-termini, in the absence or presence of FLP recombinase expression. Western blotting of developing pupal brains confirmed that the tagged proteins were stably expressed without abnormal degradation and were typically processed (*Artigiani et al., 2003*; *Ayoob et al., 2006*) (*Figure 1—figure supplement 1A*). Moreover, we did not observe any defects in ORN axon trajectory formation at 24hAPF in either *PlexB-Tag/+* or *PlexB-Tag/plexB⁻* flies, suggesting that the tag does not affect PlexB's functions (*Figure 1—figure supplement 1B,C*).

When performing immunostaining on *PlexB-Tag* brains using the routine protocol (*Wu and Luo, 2006*), we detected faint signal that could not consistently generate high-quality images, suggesting that endogenous PlexB proteins may be present at low levels in pupal brains. We thus adopted a tyramide-based signal amplification strategy to improve the signal-to-noise ratio (see Materials and methods). We first stained brains of *PlexB-Tag* flies without any FLP recombinase, in which all PlexB proteins should be tagged by FLAG but not V5. Indeed, no V5 signal was detected (*Figure 1K*), while the anti-FLAG antibody recognized many structures in 24hAPF brains, including the antennal lobe (*Figure 1I*). Consistent with the *PlexB-GAL4* labeling pattern, PlexB proteins were distributed both at the edge and inside of the antennal lobe, likely contributed by ORN axons and PN dendrites, respectively. Notably, the DL edge had stronger FLAG signal than the VM edge (*Figure 1I*), suggesting a non-uniform distribution of PlexB proteins in ORN axons. To confirm this, we examined

flies bearing both the *PlexB-Tag* and the ORN-specific *eyFLP*, in which PlexB proteins in ORNs should be tagged by V5 while those expressed by other cells should be tagged by FLAG. V5 staining revealed that DL axons indeed possessed a higher level of PlexB proteins (*Figure 1L*). In contrast to the ORN axon distribution, which had a DL/VM mean of 0.73, the ORN-specific PlexB protein distribution yielded a DL/VM mean of 1.43 (*Figure 1M*). Consistently, we observed a uniform distribution of PlexB inside the antennal lobe without the edge signals following FLAG staining (*Figure 1J*), further indicating that PlexB proteins are enriched in DL ORN axons. The contrasting patterns between *PlexB-GAL4* and *PlexB-Tag* suggest post-transcriptional or post-translational regulations of PlexB levels.

## PlexB overexpression causes VM ORN axons to shift to the DL trajectory, similar to PlexB loss

Given that PlexB functions in ORN axon trajectory formation (*Figure 1A,B*) (*Joo et al., 2013*), the uneven distribution of PlexB proteins in the DL and VM axon bundles raises the question of whether the PlexB protein level regulates trajectory choice. To test this, we overexpressed PlexB in ORNs and observed that the DL axon bundle was thickened at the expense of the VM bundle (*Figure 2A, B*), shifting the DL/VM ratio to 1.57 (*Figure 2C*). When the overexpression was intensified by elevating GAL4/UAS-mediated expression at 29°C, the DL shift of ORN axons became more severe (*Figure 2C*), indicating that elevating the PlexB level drives ORN axons to the DL trajectory in a scalable manner. We observed the same phenotype – loss of the VM trajectory – when PlexB was overexpressed only in two ORN classes labeled by *AM29-GAL4* (*Figure 4—figure supplement 1A,B*), excluding the possibility that the DL shift was an artifact caused by pan-ORN PlexB overexpression.

Notably, PlexB overexpression phenotypically resembled loss of PlexB, both of which caused a net VM-to-DL shift of ORN axons (*Figure 2C*). This observation indicates that the formation of the VM trajectory requires not only the presence of PlexB but also a specific level of it. In line with the distribution pattern of PlexB proteins (*Figure 1L*), these findings suggest that the PlexB level directs the trajectory choice of ORN axons.

In addition to the defect in trajectory choice, we also observed defasciculation of ORN axons in *plexB* mutants (*Figure 1B* and *Figure 4—figure supplement 1E,F*) (*Joo et al., 2013*), a phenotype also observed in the development of the embryonic longitudinal tract (*Ayoob et al., 2006*). However, axon defasciculation was not observed in PlexB-overexpressing ORNs at either the pupal or the adult stage. Thus, the presence of PlexB at a high or low level is sufficient to mediate axon bundling, distinct from its level-dependent effects in trajectory choice.

## Moderate levels of PlexB knockdown cause DL ORN axons to shift to the VM trajectory

To further investigate the relationship between the PlexB level and ORN axon trajectory choice, we asked if PlexB knockdown drives DL axons to the VM trajectory. However, a genetic driver specific to DL ORNs at 24hAPF is not available. Using a pan-ORN driver for both axon labeling and PlexB knockdown at 24hAPF would lead to two problems: (1) DL and VM axons are not distinguishable from each other; and (2) PlexB knockdown shifts VM axons to the DL trajectory, as in *plexB* mutants, impeding our observation of the trajectory choice made by DL axons. To unambiguously examine DL axons in PlexB knockdown, we inspected the adult brains and used *Or67d-QF* to label DA1 ORNs, whose axons predominantly choose the DL trajectory and stay at the DL edge of the adult antennal lobe (yellow arrowheads in *Figure 2D*). When PlexB was knocked down by RNA interference (RNAi) in ORNs, a subset of DA1 ORN axons innervated the VM half of the antennal lobe (red arrow in *Figure 2E*) while the DL axon bundle was substantially diminished (yellow arrowheads in *Figure 2E*), indicating that lowering the PlexB level drives DL axons to the VM trajectory. Despite starting with an incorrect trajectory far from their target glomerulus, a portion of the ectopic VM axons projected back to the DL region and correctly innervated DA1 through a medial entry point (red arrowhead in *Figure 2E*) instead of the original entry points at the edge (yellow arrowheads in *Figure 2D,E*).

In a GAL4/UAS-driven RNAi experiment, phenotypic penetrance typically increases with temperature due to elevated GAL4 activity at higher temperatures. Interestingly, the percentage of antennal lobes with ectopic VM axons dropped when PlexB knockdown was performed at higher

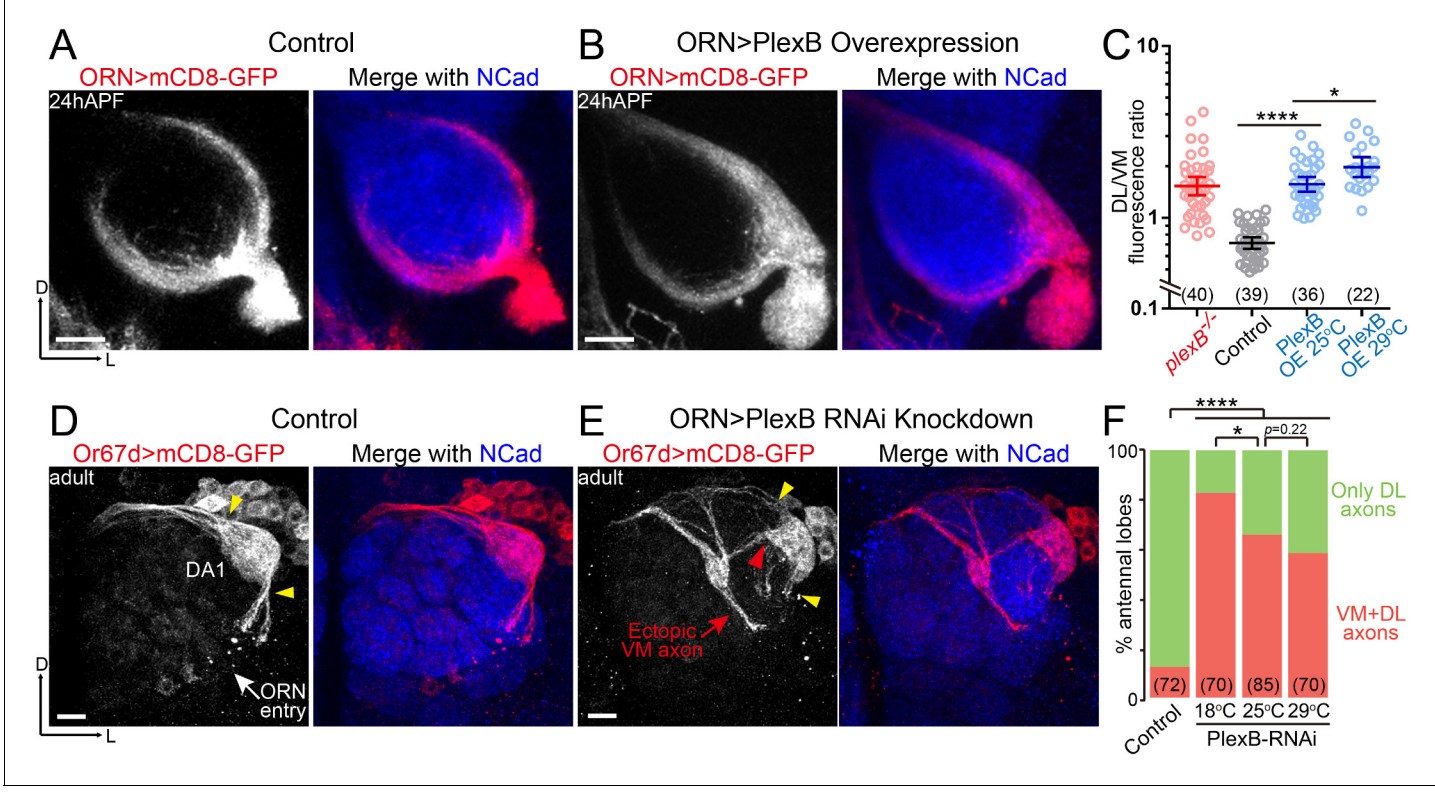

**Figure 2.** ORN axon trajectory choice requires specific levels of PlexB. (**A**) ORN axon trajectories in *wild-type* flies at 24hAPF. (**B**) Overexpression of PlexB in ORNs shifts ORN axons to the DL trajectory. (**C**) Fluorescence intensity ratios of ORN axon trajectories (DL/VM). Geometric means: *plexB⁻/⁻*, 1.53; control, 0.71; PlexB OE at 25°C, 1.57; PlexB OE at 29°C, 1.98. (**D**) ORN axons targeting to the DA1 glomerulus were labeled by membrane-tethered GFP driven by *Or67d-QF*. In *wild-type* flies, DA1 ORN axons stay only at the DL edge of the antennal lobe (yellow arrowheads). *Or67d-QF* has non-specific labeling of several non-neuronal cells outside of each antennal lobe. No axon or dendrite was observed from these cells. (**E**) PlexB RNA interference (RNAi) in ORNs leads to the formation of a VM axon bundle of DA1 ORNs (red arrow). A subset of ectopic VM axons innervates the DA1 glomerulus at a medial entry point (red arrowhead), instead of the normal entry points at the edge of the antennal lobe (yellow arrowheads). (**F**) Quantification of antennal lobes with the VM axon bundle of DA1 ORNs in *wild-type* flies and PlexB RNAi flies at different temperatures. Sample sizes are noted in parentheses. Significance among multiple groups in *Figure 2C* was determined by one-way ANOVA with Tukey's test for multiple comparisons. Significance of the contingency table in *Figure 2F* was determined by Fisher's exact test. *p<0.05; ****p<0.0001. Images are shown as maximum z-projections of confocal stacks. Scale bars, 10 µm. Axes, D (dorsal), L (lateral).
DOI: https://doi.org/10.7554/eLife.39088.004

temperatures (*Figure 2F*). Considering the VM-to-DL trajectory shift observed in *plexB* mutants, it is plausible that a severe knockdown of PlexB (e.g., at 29°C) resembled the *plexB* mutant, in which the DL targeting of DA1 axons was caused by the loss of PlexB. Consistently, all DA1 axons stay DL in *plexB* mutants (*Joo et al., 2013*). In line with the low level, but not absence, of PlexB proteins in the VM trajectory (*Figure 1L*), these observations further support the notion that the VM trajectory requires an intermediate PlexB level.

## Differential distribution of PlexB proteins in ORN axons at the stage of glomerular selection

Structural and gene expression dynamics occur throughout the assembly of *Drosophila* olfactory circuits (*Jefferis et al., 2004*; *Li et al., 2017*). At 24hAPF, ORN axons form two trajectories circumscribing the antennal lobe before entering it (*Figure 3A*). Within the next 24 hr, ORN axons innervate the antennal lobe, arborize in their targeting territories, and search for the dendrites of their PN partners. By 48hAPF, the antennal lobe is significantly enlarged and houses proto-glomeruli where the axons and dendrites of matching ORNs and PNs interact (*Figure 3B*). We wondered if PlexB continuously impacts ORN axon wiring after instructing trajectory choice. To address this question, we examined PlexB expression at 48hAPF using *PlexB-GAL4* and *PlexB-Tag*.

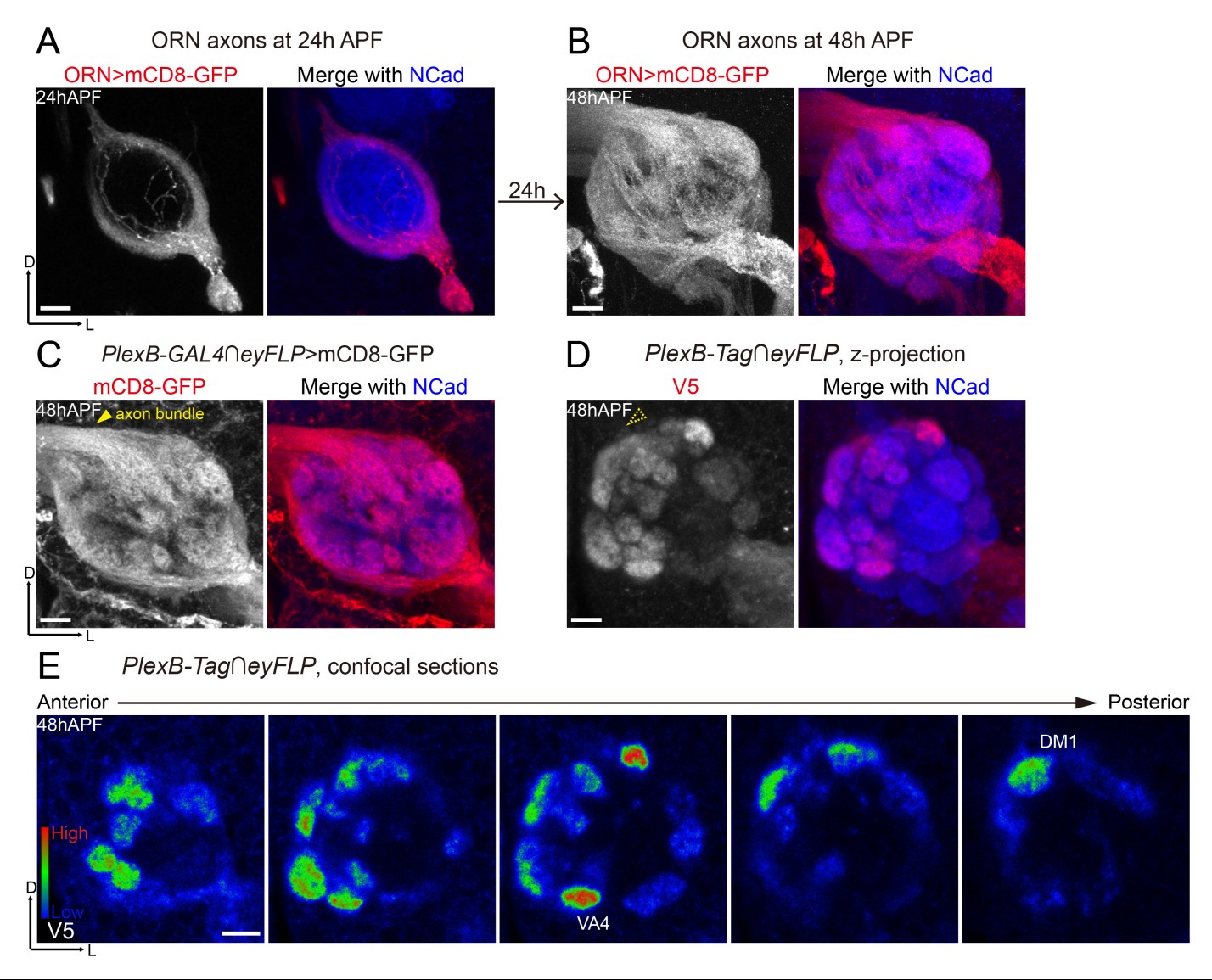

**Figure 3.** Differential distribution of PlexB proteins in ORN axons at the stage of glomerular selection. (**A**) At 24hAPF, ORN axons bifurcate and form the DL and VM trajectories circumscribing the antennal lobe. ORN axons were labeled by the pan-ORN *Peb-GAL4* driver line (*Sweeney et al., 2007*). (**B**) By 48hAPF, ORN axons have innervated the antennal lobe, selected targeting areas, and formed proto-glomeruli with prospective PN partners. Within 24 hr (24hAPF to 48hAPF), the antennal lobe is significantly enlarged and ORN axons have extensively elaborated terminal arbors. (**C**) Intersection of *PlexB-GAL4* and *eyFLP* (FLP in ORNs) labels nearly all glomeruli. The generic membrane-anchored mCD8-GFP uniformly labels axons, including the commissure bundle (arrowhead). (**D, E**) At 48hAPF, ORN axons targeting to several discrete glomeruli express higher levels of PlexB proteins than their neighbors. These PlexB-high glomeruli include VA4, DM1, and a dorsal one whose identity cannot be unambiguously determined at this stage. (**D**) Maximum z-projection of a confocal stack. Glomerular boundaries are visible while the commissure axon bundle is barely observed (dotted arrowhead). (**E**) Single confocal sections from anterior to posterior. Colors represent the intensity of V5 signal: red, high; green, intermediate; blue, low. Images are shown as maximum z-projections of confocal stacks, except for *Figure 3E*, where single confocal sections are shown. Scale bars, 10 μm. Axes, D (dorsal), L (lateral).

DOI: https://doi.org/10.7554/eLife.39088.005

The following figure supplement is available for figure 3:

**Figure supplement 1.** *PlexB-GAL4* labels almost all ORNs at 48hAPF, and *PlexB* proteins in ORN axons undergo dynamic changes between 30hAPF and 36hAPF.

DOI: https://doi.org/10.7554/eLife.39088.006

When intersected with the ORN-specific *eyFLP*, *PlexB-GAL4* labeled ORN axons of almost every glomerulus at 48hAPF (*Figure 3C*), consistent with the observation that *PlexB-GAL4* labeled almost all ORN somas in the antenna (*Figure 3C*, *Figure 3—figure supplement 1A*). However, PlexB proteins were not uniformly distributed in all ORN axons (*Figure 3D*), further indicating the presence of post-transcriptional or post-translational regulations of PlexB expression. Among ORN axons, the ones targeting to the medial glomeruli were the main contributors of PlexB proteins at 48hAPF (*Figure 3D*), producing a PlexB distribution pattern (medial high and lateral low) distinct from that of 24hAPF (DL high and VM low) through a rapid transition between 30 hr and 36hAPF (*Figure 3—figure supplement 1B,C*). Notably, several discrete glomeruli had significantly higher ORN PlexB levels than their neighbors (*Figure 3E*), including VA4, DM1, and another dorsal glomerulus whose identity could not be unambiguously determined by its position and morphology at 48hAPF.

In contrast to concentration in the axons at 24hAPF (*Figure 1L*), the ORN-specific PlexB proteins were highly enriched in the neuropil at 48hAPF (*Figure 3D*), where ORN axons meet PN dendrites for partner selection. As shown in *Figure 3D*, the glomerular boundaries were visible while the commissure axon bundle was barely detectable (dotted arrowhead in *Figure 3D*), in contrast to the uniform axonal labeling by the generic membrane-anchored mCD8-GFP (*Figure 3C*). Thus, besides the global change of expression pattern, the subcellular localization of PlexB proteins is also altered from 24hAPF to 48hAPF.

This differential and neuropil-enriched distribution of PlexB proteins at the stage of ORN glomerular selection prompted us to investigate if the PlexB level regulates glomerular targeting in addition to early-stage trajectory choice.

## PlexB protein level regulates glomerular selection independently of trajectory choice

Examining ORN glomerular targeting following manipulation of the PlexB level not only allows us to test if the PlexB level regulates glomerular selection, but also provides a means to answer a long-standing question: how does a preceding developmental event (i.e., trajectory choice at 24hAPF) affect a subsequent process (i.e., glomerular selection at 48hAPF) in the assembly of a circuit? To this end, we characterized the glomerular targeting, while manipulating the PlexB level, of two sets of ORNs: (1) Or92a+ ORNs, whose axons normally choose the VM trajectory and innervate the VA2 glomerulus (*Figure 4A*); and (2) AM29+ ORNs, which include two classes of ORNs targeting to the DL4 and DM6 glomeruli via the DL and VM trajectories, respectively (*Figure 4—figure supplement 1A*). Although *AM29-GAL4* provides specific access to two ORN classes, its expression is too late and weak to drive RNAi to reduce the PlexB level, impelling us to use the *plexB* mutant instead.

Consistent with the previous observations at 24hAPF (*Figure 2C*), both overexpression and loss of PlexB drove axons of Or92a+ ORNs to the DL side (*Figure 4B,E*). In the case of AM29+ ORNs, the VM axon bundle disappeared in both conditions and likely merged with the original DL track (*Figure 4—figure supplement 1B,E*). As a consequence, most ectopic axons innervated ectopic glomeruli in the dorsal half of the antennal lobe (*Figure 4C,F* and *Figure 4—figure supplement 1C,F*). The preceding step of trajectory choice thus largely constrained the possibilities of subsequent glomerular selection by dividing ORN axons into two spatially separated populations. However, we also observed a small fraction of ectopic axons that ultimately projected back to the VM area (red arrowheads in *Figure 4B,E*), suggesting that ORN axon termini can explore a large area during glomerular selection to partially overcome a prior error in trajectory choice.

To test whether PlexB plays a role in target selection, we first examined glomerular choice in ectopic territory in the dorsolateral half of the antennal lobe. In contrast to their similar effects on trajectory choice, overexpression and loss of PlexB led to distinct preferences in glomerular selection. In the PlexB overexpression flies, Or92a+ ORNs mainly targeted to the PlexB-high DM1 glomerulus (*Figure 4C*), while they only chose lateral glomeruli but not DM1 in PlexB knockdown flies (*Figure 4F*). Similarly, AM29+ ORNs mis-targeted to DM1 when the PlexB level was elevated (*Figure 4—figure supplement 1C*), but mis-targeted to DM3 in *plexB* mutants (*Figure 4—figure supplement 1E,F,G*). Thus, the PlexB level biases glomerular selection when ORNs are forced to ectopically innervate due to incorrect trajectory.

We then asked if the PlexB level also regulates glomerular targeting of ORNs near their original territories. The time window between trajectory choice and glomerular selection was too short to allow a robust genetic switch, such as temperature-controlled *GAL80*, to manipulate the level of

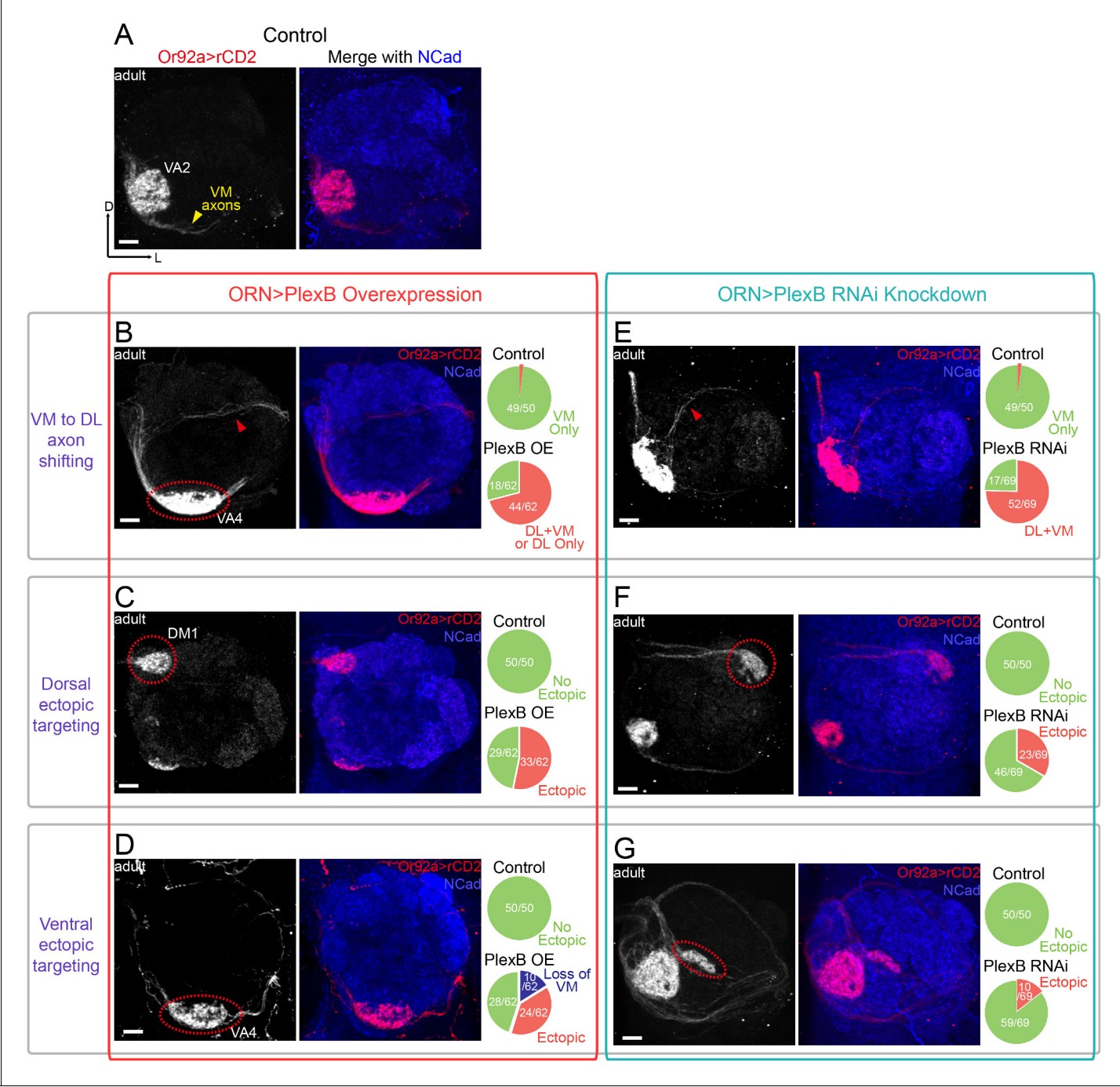

**Figure 4.** ORN glomerular selection requires specific levels of PlexB. (A) ORN axons targeting to the VA2 glomerulus were labeled by membrane-localized rCD2 driven by an *Or92a* promoter. Or92a+ ORN axons predominantly choose the VM trajectory (yellow arrowhead). (B) Some Or92a+ ORN axons choose the DL trajectory (red arrowhead) in the PlexB overexpression flies. In the shown antennal lobe, Or92a+ ORN axons targeting to the VA4 glomerulus are also observed (dotted red circle). (C) PlexB overexpression causes Or92a+ ORN axons to ectopically target to dorsal glomeruli (dotted red circle), mainly DM1 (67% of dorsal mis-targeting cases). (D) PlexB overexpression causes Or92a+ ORN axons to ectopically innervate ventral glomeruli (dotted red circle), mainly VA4 (88% of ventral mis-targeting cases). (E) Some Or92a+ ORN axons choose the DL trajectory (red arrowhead) when PlexB is knocked down by RNAi in ORNs. (F) PlexB knockdown causes Or92a+ ORN axons to incorrectly target to dorsolateral glomeruli (dotted red circle), distinct from the stereotyped DM1 mis-targeting in the PlexB overexpression flies. (G) In a few PlexB knockdown flies, local ectopic targeting near VA2 was observed (dotted red circle). However, mis-targeting to VA4, a predominant phenotype in PlexB overexpression, was not observed. Antennal lobe counts of each phenotype and sample sizes are noted in the pie charts. Images are shown as maximum z-projections of confocal stacks. Scale bars, 10 μm. Axes, D (dorsal), L (lateral).

*Figure 4 continued on next page*

*Figure 4 continued*

DOI: https://doi.org/10.7554/eLife.39088.007
The following figure supplement is available for figure 4:

**Figure supplement 1.** Glomerular selection of AM29+ ORNs requires specific levels of PlexB.
DOI: https://doi.org/10.7554/eLife.39088.008

PlexB. Because overexpression or loss of PlexB only shifted a fraction of ORN axons to the DL trajectory, most antennal lobes still had axons in the correct VM trajectory; this provided a means to probe the trajectory-independent function of PlexB. We found that PlexB overexpression led to stereotyped mis-targeting of Or92a+ ORNs to the nearby, PlexB-high VA4 glomerulus, along with the loss of VA2 innervation (*Figure 4B,D*). PlexB knockdown also caused aberrant targeting around the VA2 area in a small fraction of antennal lobes (*Figure 4G*), but no innervation to VA4 was observed in the 69 antennal lobes examined. In more than half of the antennal lobes of PlexB overexpression flies, the AM29+ ORNs had a normal VM axon track but lost DM6 innervation (*Figure 4—figure supplement 1D*), which was also observed in a few *plexB* mutant flies (*Figure 4—figure supplement 1G*).

Notably, the distinct targeting preferences caused by overexpression or loss of PlexB match the endogenous PlexB distribution. Elevating the PlexB level predominantly drove axons to VA4 or DM1, both of which exhibited high endogenous PlexB levels (*Figure 3E*). PlexB knockdown and mutants caused aberrant targeting to DM3 or the lateral glomeruli, whose PlexB levels were low (*Figure 3D,E*), but not to the PlexB-high glomeruli such as VA4. Thus, independently of its role in trajectory choice, the PlexB level also regulates subsequent glomerular selection in the assembly of the olfactory circuit.

## Sema2b antagonizes the effect of high-level PlexB

We previously observed that Sema2b, the canonical ligand of PlexB, is highly enriched in the VM bundle of ORN axons (*Figure 5—figure supplement 1A*) (*Joo et al., 2013*), while PlexB proteins are more abundant in DL axons (*Figure 1L*). Moreover, loss of Sema2b causes ORN axons to shift to the DL trajectory (*Figure 5H*) (*Joo et al., 2013*), similar to PlexB overexpression (*Figure 2B*). Along with the opposing distribution, this phenotypic similarity between loss of Sema2b and PlexB overexpression suggests a potentially antagonistic relationship between Sema2b and PlexB in instructing ORN trajectory choice. We therefore tested genetic interactions between *Sema2b* and *PlexB*.

Unlike the homozygotes of the *sema2b* mutant (*Figure 5H*), heterozygous flies had normal ORN axon trajectories, indistinguishable from those of *wild-type* flies (*Figure 5A,B,E*). However, the heterozygosity of *sema2b* significantly promoted the VM-to-DL axon shift caused by elevating the PlexB level (*Figure 5C,D,E*). Although PlexB overexpression drove axons to the DL trajectory, the VM bundle was visible in every antennal lobe in a *wild-type* background (*Figure 5C*). In a subset of *sema2b* heterozygotes, however, elevating the PlexB level in ORNs almost completely eliminated the VM trajectory, leading to a single, thick DL bundle (*Figure 5D*). In line with this observation at 24hAPF, *sema2b* heterozygosity also enhanced the VM-to-DL shift of Or92a+ ORN axons caused by PlexB overexpression when examined at the adult stage (*Figure 5K* and *Figure 5—figure supplement 1B*). Thus, loss of one-copy of the *Sema2b* gene boosts the effect of PlexB overexpression, suggesting that Sema2b partially 'blocks' the effect of high-level PlexB in *wild-type* flies. Moreover, the phenotypic expression of *sema2b* heterozygosity relied on an elevated PlexB level, suggesting that Sema2b controls ORN axon trajectory choice partly by modulating the level-dependent effects of PlexB.

To further delineate the interactions of Sema2b and PlexB, we tested if changing the PlexB level modifies the VM-to-DL trajectory shift in *sema2b* mutants (*Figure 5H*). Examining trajectory choice when elevating the PlexB level in *sema2b* homozygotes would not be informative since they individually lead to the same consequence – VM-to-DL shift ORN axons and loss of one copy of the *Sema2b* gene has shown phenotypic enhancement to PlexB overexpression. We thus tested if lowering the PlexB level alters the trajectory phenotype in s*ema2b* mutants. PlexB knockdown in a *wild-type* background caused a slight DL shift of ORN axons (*Figure 5F,G,J*), similar to the loss of PlexB but to a lesser extent. While loss of Sema2b also led to a DL shift of ORN axons, the combination of PlexB

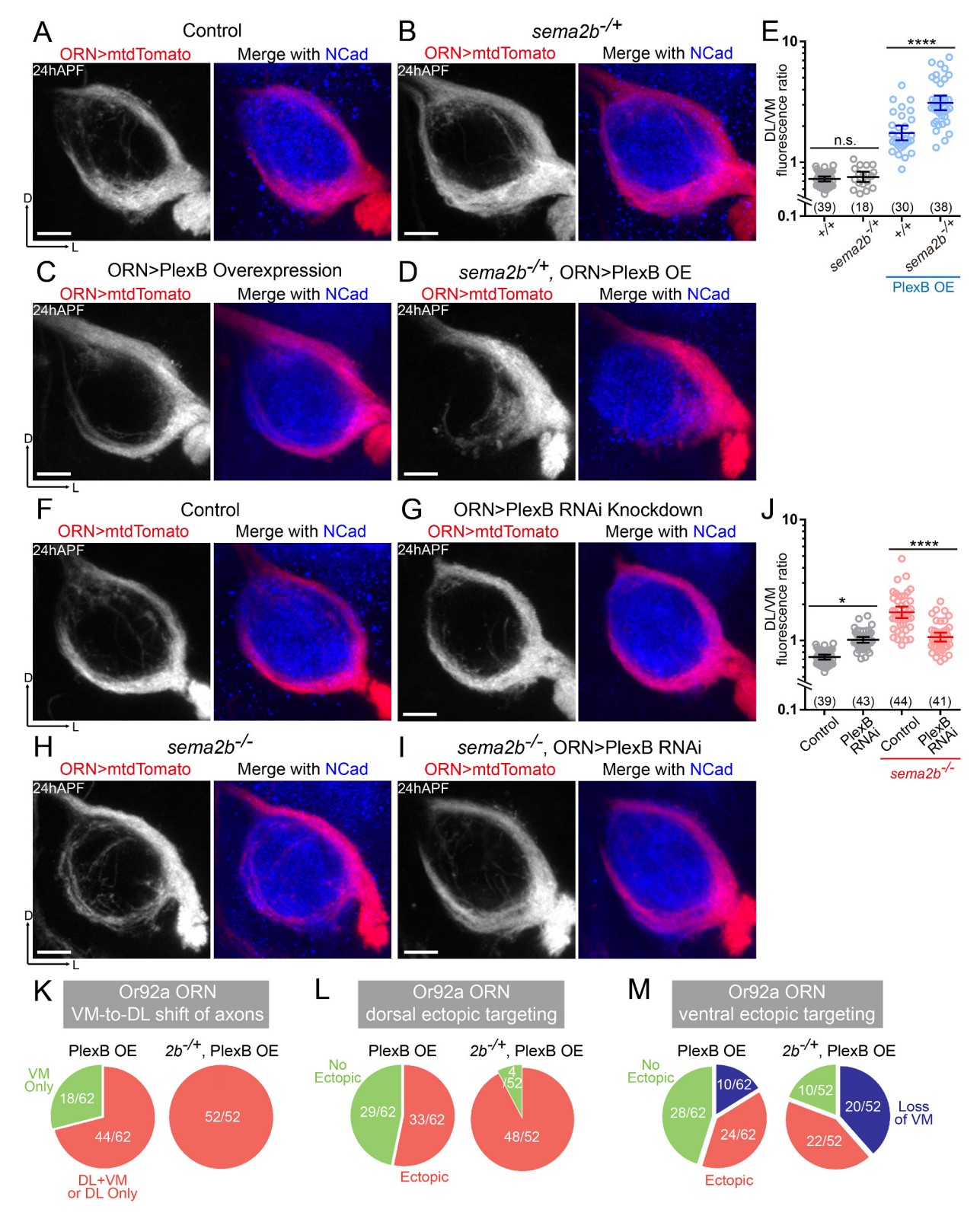

**Figure 5.** Sema2b antagonizes the effect of high-level PlexB. (**A**) ORN axon trajectories of *wild-type* flies at 24hAPF. (**B**) *sema2b* heterozygous flies have normal ORN axon trajectories. (**C**) PlexB overexpression shifts ORN axons to the DL trajectory. (**D**) *sema2b* heterozygosity (*sema2b$^{-/+}$*) enhances the DL shift of ORN axons caused by PlexB overexpression. (**E**) Fluorescence intensity ratios of ORN axon trajectories (DL/VM). Geometric means: *+/+*, 0.73; *sema2b$^{-/+}$*, 0.76; PlexB OE in *+/+*, 1.75; PlexB OE in *sema2b$^{-/+}$*, 3.10. (**F**) ORN axon trajectories of *wild-type* flies at 24hAPF. (**G**) PlexB knockdown by

*Figure 5 continued on next page*

*Figure 5 continued*

RNA interference (RNAi) in ORNs slightly shifts ORN axons to the DL trajectory, phenotypically resembling *plexB* mutants but with less severity. (H) In *sema2b* mutant flies, ORN axons preferentially choose the DL trajectory at 24hAPF. (I) PlexB RNAi in ORNs suppresses the DL shift of ORN axons caused by the loss of Sema2b. (J) Fluorescence intensity ratios of ORN axon trajectories (DL/VM). Geometric means: control, 0.73; PlexB RNAi, 1.01; *sema2b*$^{-/-}$, 1.83; PlexB RNAi in *sema2b*$^{-/-}$, 1.11. (K, L, M) *sema2b* heterozygosity (*sema2b*$^{-/+}$) enhances both incorrect axon trajectory choice and ectopic glomerular targeting caused by PlexB overexpression. Each phenotype is described in detail in *Figure 4*. (K) VM-to-DL shift of ORN axons, as in *Figure 4B*; (L) dorsal ectopic targeting, as in *Figure 4C*; (M) ventral ectopic targeting, as in *Figure 4D*. Sample sizes are noted in parentheses. Antennal lobe counts of each phenotype and sample sizes are noted in the pie charts. Significance among multiple groups was determined by one-way ANOVA with Tukey's test for multiple comparisons. n.s., not significant; *$p < 0.05$; ****$p < 0.0001$. Images are shown as maximum z-projections of confocal stacks. Scale bars, 10 µm. Axes, D (dorsal), L (lateral).

DOI: https://doi.org/10.7554/eLife.39088.009

The following figure supplement is available for figure 5:

**Figure supplement 1.**

DOI: https://doi.org/10.7554/eLife.39088.010

knockdown and *sema2b* mutant did not drive more axons to the DL bundle; rather, it suppressed the *sema2b* mutant-induced DL shift (*Figure 5H,I*), yielding a DL/VM mean of 1.11 in contrast to 1.83 of *sema2b* mutants (*Figure 5J*). Thus, lowering the PlexB level can, at least partially, substitute the function of Sema2b in ORN trajectory choice, in line with the notion that Sema2b antagonizes high-level PlexB.

In addition to regulating ORN trajectory choice at 24hAPF, specific levels of PlexB are required for subsequent glomerular selection. By examining the adult glomerular targeting of Or92a+ and AM29+ ORNs, we observed more antennal lobes with incorrect glomerular innervation when overexpressing PlexB in *sema2b* heterozygotes (*Figure 5L,M* and *Figure 5—figure supplement 1C,D*), suggesting that Sema2b also antagonizes the effect of high-level PlexB in glomerular selection.

## Discussion

Here, we show that PlexB proteins are distributed at different levels in developing ORN axons, and these patterns change dynamically during development (*Figure 1* and *Figure 3*). Accordingly, the level of PlexB consecutively controls two critical steps in the assembly of the olfactory map – ORN axon trajectory choice and glomerular selection (*Figure 2* and *Figure 4*). Sema2b, a partner of PlexB, antagonizes the effect of high-level PlexB to further ensure the wiring fidelity (*Figure 5*). Thus, a single guidance molecule, PlexB, acts in a multi-step developmental process by stage-specific expression and level-dependent effects (*Figure 6*).

### PlexB expression and protein distribution in developing olfactory circuits

Spatially and temporally appropriate distribution of a protein is a prerequisite for its normal function, particularly in developmental processes with structural and gene expression changes. Thus, uncovering the expression pattern of a protein at high spatiotemporal resolution provides essential information for dissecting its functions. Moreover, it is vital to resolve the protein distribution in a genetically defined cell population, as different types of cells, especially in the nervous system, are commonly intermingled in vivo, potentially obscuring the observation and interpretation. To reveal the distribution of endogenous PlexB proteins specifically in ORN axons, we devised a FLP-gated conditional tagging strategy (*Figure 1H*) that preserved most, if not all, endogenous regulatory components of the *PlexB* gene. Indeed, this cell-type specific tagging increased the resolution (comparison between *Figure 1I and L*) and revealed that PlexB proteins are more enriched in DL axons compared to VM axons at 24hAPF while expressed at high levels in several discrete glomeruli at 48hAPF (*Figure 6A*), demonstrating a rapid transition of PlexB protein distribution within 24 hr (*Figure 1L*, *Figure 3—figure supplement 1B,C*, and *Figure 3D*). We further found that the PlexB level controls ORN axon trajectory choice and glomerular selection at these two stages, respectively. Thus, through rapid reshaping of its distribution, PlexB instructs two consecutive developmental steps.

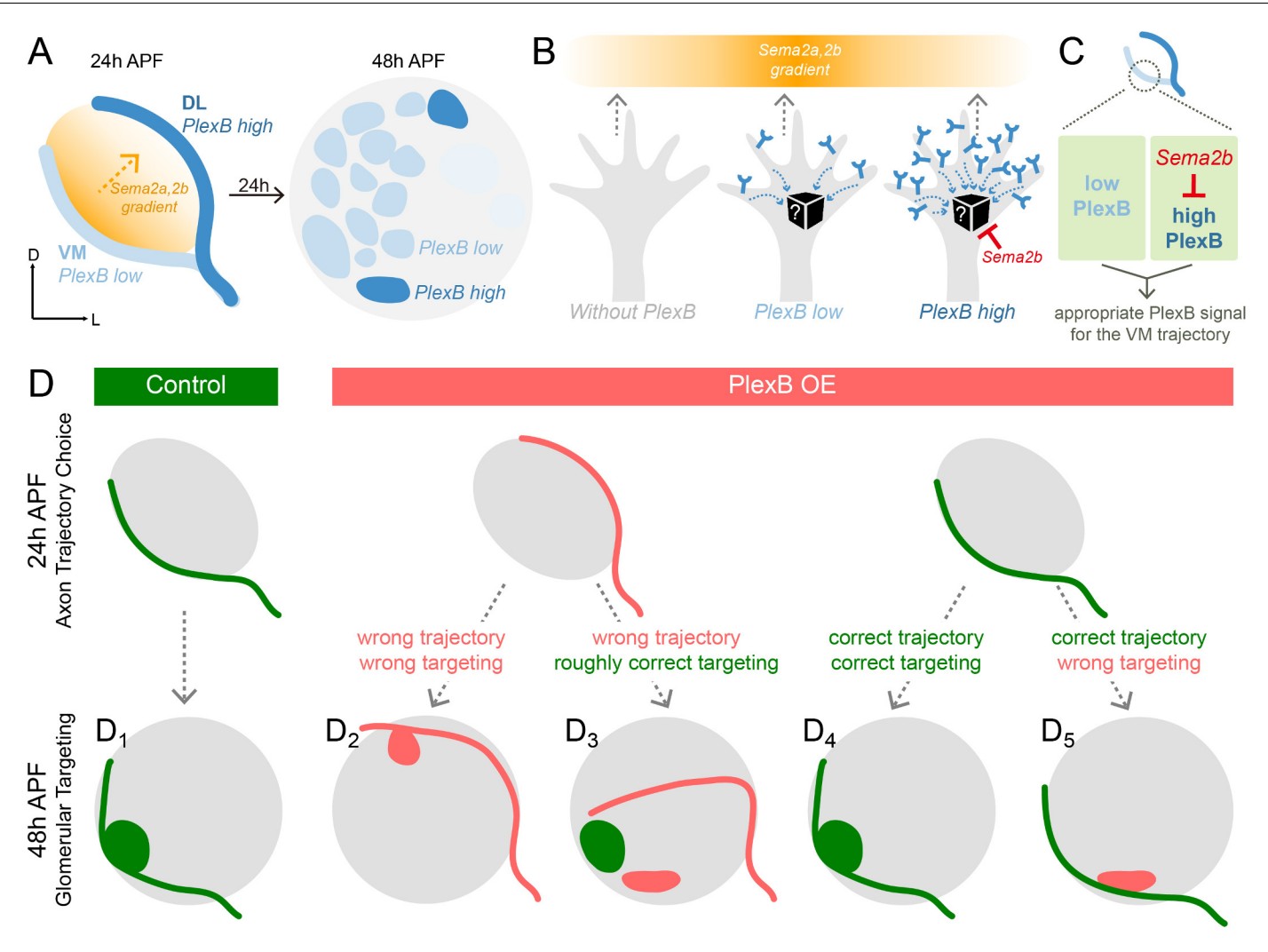

**Figure 6.** Schematic summary of the roles of PlexB in the stepwise assembly of the *Drosophila* olfactory circuit. (A) At 24hAPF, ORN axons of the DL trajectory express a higher level of PlexB proteins, compared to the VM axons. Sema2a and Sema2b proteins form a descending gradient from VM to DL in the antennal lobe (*Joo et al., 2013*). Within the next 24 hr, ORN axons enter the antennal lobe and form proto-glomeruli with PN dendrites. At 48hAPF, ORN axons innervating several discrete glomeruli express PlexB proteins at high levels, exhibiting a different distribution pattern from that of 24hAPF. (B) ORN axons without PlexB (as in the *plexB* mutant) or with high-level PlexB (as the *wild-type* DL axons or the VM axons in PlexB overexpression) preferentially choose the low-Sema2a/2b area (DL). Only ORN axons with low-level PlexB (as the *wild-type* VM axons or the DL axons in PlexB knockdown) target to the high-Sema2a/2b area (VM). Although it is unknown how PlexB levels affect its signaling (black box), Sema2b antagonizes the activity of high-level PlexB. (C) ORN axons destined to the VM trajectory express low levels of PlexB. Additionally, Sema2b is enriched in the VM bundle to antagonize the signal of high-level PlexB. These two mechanisms thus ensure an appropriate PlexB signal for the formation of the VM trajectory. (D) ORN glomerular selection is affected by, but also can be independent of, the preceding trajectory choice. Although ectopic glomerular targeting frequently occurs when the trajectory goes wrong ($D_2$), a portion of ORN axons can overcome the initial trajectory choice and return to the roughly correct area ($D_3$). Among ORN axons achieving correct trajectory choice, many innervate nearby incorrect glomeruli if the PlexB level is disturbed ($D_5$).

DOI: https://doi.org/10.7554/eLife.39088.011

We also examined a transcriptional reporter of PlexB (*Figure 1D*) and observed that it labeled most ORNs at both 24hAPF and 48hAPF (*Figure 1F* and *Figure 3C*). Although the perdurance of *mCD8-GFP* may raise the concern of whether it can faithfully report the dynamics of *PlexB* transcription, the massive expansion of ORN axonal surface area in the developing antennal lobe (*Figure 3A, B*) likely dilutes mCD8-GFP carried over from earlier stages; thus the signal we detected was likely contributed mainly by recently translated mCD8-GFP proteins. Compared to the broad labeling of

*PlexB-GAL4*, the more restricted distribution of PlexB proteins suggests that the expression of PlexB is subject to post-transcriptional and/or post-translational regulations. In contrast to the long time-scales underlying transcriptional regulations, the molecular mechanisms directly controlling RNA or protein dynamics enable faster turnover, which is better suited for the rapid reshaping of PlexB protein distribution in developing olfactory circuits. The exact mechanism by which PlexB levels are post-transcriptionally regulated requires future investigations.

## Level-dependent effects of PlexB

Our data indicate that just the presence of PlexB is insufficient for the precise assembly of the *Drosophila* olfactory map. Analysis of ORN axon trajectory choice (*Figure 2*) and glomerular selection (*Figure 4*) revealed that each class of ORNs requires a specific level of PlexB at each developmental step for appropriate wiring. We further note that PlexB overexpression (*Hu et al., 2001*) and *plexB* mutant (*Ayoob et al., 2006*) cause similar innervation defects in embryonic ISNb motor axons, indicating that specific PlexB levels may be commonly required in the development of the nervous system.

In the formation of ORN axon trajectories, VM axons need a specific level of PlexB to make the correct choice, as both overexpression and loss of PlexB shifted them to the DL trajectory. We previously found that the canonical ligands of PlexB, Sema2a and Sema2b, establish a VM-high, DL-low gradient in the antennal lobe at 24hAPF (*Figure 6A*) that specifies ORN axon trajectory choice (*Joo et al., 2013*). Thus, axons with a high level of PlexB avoid regions with high levels of Sema2a/2b, whereas the PlexB-low axons appear to be attracted to regions with high levels of Sema2a/2b (*Figure 6B*). This observation suggests that different PlexB levels induce divergent behaviors of growth cones in response to external semaphorin cues, raising the question on the molecular basis of this level-dependent PlexB signaling (black box in *Figure 6B*). Although Rac and RhoA have been shown to mediate PlexB signaling in vivo (*Hu et al., 2001*), a direct and real-time reporter of PlexB's molecular activity has not been developed. We anticipate that such a PlexB signal reporter, along with a better biophysical understanding of Plexins (*Pascoe et al., 2015*; *Seiradake et al., 2016*; *Siebold and Jones, 2013*), could reveal the level-dependent signaling of PlexB in the future. Furthermore, live imaging of axon dynamics during circuit assembly (*Langen et al., 2015*; *Özel et al., 2015*) and genetic toolkits allowing mosaic analysis of fourth-chromosome genes of *Drosophila* (e.g., *PlexB*) will uncover how level-dependent PlexB signal controls the behavior of single axons, complementing to our current analyses of bulk axons in fixed tissues.

In addition to its canonical role as a secreted guidance molecule, we discovered that Sema2b antagonizes the activity of high-level PlexB in developing olfactory circuits (*Figure 5*). During ORN axon trajectory choice, Sema2b proteins are highly enriched in the VM bundle. Our genetic analysis identified two mechanisms ensuring the correct formation of the VM trajectory: (1) ORN axons destined to the VM bundle express lower levels of PlexB; and (2) Sema2b in VM axons antagonizes the activity of high-level PlexB (*Figure 6C*). One plausible mechanism underlying this antagonistic effect is that co-expression of Sema2b desensitizes most PlexB receptors in VM axons, as observed in other Plexins and GPCRs (*Gainetdinov et al., 2004*; *Piper et al., 2005*). This antagonistic effect may contribute to our previous observation that Sema2b, a secreted protein, can play a cell-autonomous function in ORN trajectory choice (*Joo et al., 2013*). Exploring this possibility further necessitates an in vitro PlexB signaling assay complementary to the in vivo developmental readout, like the growth cone collapsing test used to study Sema3A and its receptors (*Luo et al., 1993*), to dissect the molecular activity of PlexB.

## Stepwise assembly of the olfactory map

Building a sophisticated, error-free neural network requires many temporally coordinated steps. To elucidate how a preceding event affects a subsequent process, we examined two consecutive steps in ORN axon guidance – trajectory choice and glomerular targeting, both of which utilize PlexB levels to instruct the wiring process (*Figure 6D*). In a *wild-type* fly, the early trajectory choice guides axons to the proximity of their destined targeting areas and thus facilitates the next step, glomerular selection (*Figure 6D$_1$*). When trajectory choice is disrupted, as when PlexB is overexpressed, most axons aberrantly arborize in the territory of the altered trajectory, indicating that the previous step constrains the possibilities of following selections (*Figure 6D$_2$*). However, in a few cases, axons

initially on the wrong track are able to project back to the roughly correct targeting area, indicating that axons also explore a wide area for glomerular selection (*Figure 6D$_3$*). Moreover, correct trajectory choice does not necessarily lead to appropriate glomerular innervation in flies with altered PlexB levels, as glomerular choice is also regulated by the level of PlexB (*Figure 6D$_4$,D$_5$*).

By dividing the antennal lobe into two halves in the first step, this stepwise wiring strategy reduces the possibilities and thus the complexity of subsequent glomerular selection, endowing higher robustness to the developmental program (*Hong and Luo, 2014*). Importantly, this strategy also allows the reuse of a molecule, such as PlexB, for distinct purposes in multiple steps to maximize its capacity. Coupling this stepwise wiring strategy with the functional versatility of individual molecules thus contributes to solving the conflict between the limited number of guidance molecules and the complexity of the neural network.

# Materials and methods

## Key resources table

| Designation | Source or reference | Identifiers | Additional information |
|---|---|---|---|
| *Pebbled-GAL4* | doi:10.1016/j.neuron.2006.12.022 | | |
| *AM29-GAL4* | doi:10.1038/nn1832 | | |
| *Or67d-QF* | doi:10.1016/j.neuron.2013.06.014 | | |
| *Or92a-rCD2* | this study | | Or92a promoter fused to a rat CD2 coding sequence in a P-element vector |
| *eyFLP$^{3.5}$* | doi:10.1016/j.neuron.2005.09.019 | | |
| *UAS-mtdTomato* | doi:10.1016/j.cell.2010.02.025 | RRID:BDSC_30124 | |
| *UAS-mCD8-GFP* | doi:10.1016/S0896-6273 (00)80701–1 | RRID:BDSC_5130 | |
| *UAS-FRT-Stop-FRT-mCD8-GFP* | doi:10.1038/nn.2442 | RRID:BDSC_30125 | |
| *QUAS-mCD8-GFP* | doi:10.1016/j.cell.2010.02.025 | RRID:BDSC_30002 | |
| *UAS-RedStinger* | doi:10.2144/04363ST03 | RRID:BDSC_8547 | |
| *plexB$^{KG00878}$* | doi:10.1534/genetics.104.026427 | RRID:BDSC_14579 | |
| *PlexB$^{MI15559}$* | doi:10.1038/nmeth.1662 | RRID:BDSC_61730 | |
| *PlexB-Tag* | this study | | Endogenous conditional tagging of PlexB |
| *PlexB-GAL4* | Xiaojun Xie and Alex Kolodkin, unpublished | | Coding intron MiMIC-T2A-GAL4 of PlexB |
| *UAS-PlexB-RNAi* | doi:10.1038/nmeth.1592 | RRID:BDSC_57813 | |
| *UAS-PlexB* | doi:10.1016/j.neuron.2013.03.022 | | |
| *sema2b$^{f02042}$* | doi:10.1038/ng1314 | RRID:BDSC_18505 | |
| *UAS-Sema2b* | doi:10.1016/j.neuron.2011.02.050 | | |
| rat anti-Ncad | Developmental Studies Hybridoma Bank | RRID:AB_528121 | 1:40 in 5% normal donkey serum |
| mouse anti-ELAV | Developmental Studies Hybridoma Bank | RRID:AB_528217 | 1:40 in 5% normal donkey serum |
| chicken anti-GFP | Aves Labs | RRID:AB_10000240 | 1:1000 in 5% normal donkey serum |
| rabbit anti-DsRed | Clontech | RRID:AB_10013483 | 1:200 in 5% normal donkey serum |
| mouse anti-rat CD2 | Bio-Rad | RRID:AB_321238 | 1:200 in 5% normal donkey serum |
| mouse anit-FLAG | Sigma-Aldrich | RRID:AB_439685 | 1:100 in 5% normal donkey serum |
| mouse anti-V5 | Thermo Fisher | RRID:AB_2556564 | 1:100 in 5% normal donkey serum |
| rabbit anti-Sema2b | doi:10.1016/j.neuron.2011.09.026 | | 1:500 in 5% normal donkey serum |
| ZEN | Carl Zeiss | RRID:SCR_013672 | |
| ImageJ | National Institutes of Health | RRID:SCR_003070 | |
| Prism | GraphPad | RRID:SCR_002798 | |
| Photoshop | Adobe | RRID:SCR_014199 | |
| Illustrator | Adobe | RRID:SCR_010279 | |

### *Drosophila* stocks and genotypes

Flies were raised on standard cornmeal medium with a 12 hr/12 hr light cycle at 25°C, unless otherwise specified (e.g., 29°C for enhanced transgenic expression). The following lines were used: *Pebbled-GAL4* (*Peb-GAL4*, all ORNs) (*Sweeney et al., 2007*), *AM29-GAL4* (DL4 and DM6 ORNs) (*Endo et al., 2007*), *Or67d-QF* (*Liang et al., 2013*), *Or92a-rCD2* (this study), *eyFLP*[3.5] (FLP in ORNs) (*Chotard et al., 2005*), *UAS-mtdTomato* (*Potter et al., 2010*), *UAS-mCD8-GFP* (*Lee and Luo, 1999*), *UAS-FRT-Stop-FRT-mCD8-GFP* (*Hong et al., 2009*), *QUAS-mCD8-GFP* (*Potter et al., 2010*), *UAS-RedStinger* (NLS-DsRed) (*Barolo et al., 2004*), *plexB*[KG00878] (PlexB mutant, noted as *plexB*[−]) (*Ayoob et al., 2006*; *Bellen et al., 2004*), *PlexB*[MI15559] (*Venken et al., 2011*), *PlexB-Tag* (this study), *PlexB-GAL4* (Xiaojun Xie and Alex Kolodkin, unpublished), *UAS-PlexB-RNAi* (TRiP.HMJ21821) (*Ni et al., 2011*), *UAS-PlexB* (*Joo et al., 2013*), *sema2b*[f02042] (Sema2b mutant, noted as *sema2b*[−]) (*Thibault et al., 2004*; *Wu et al., 2011*), *UAS-Sema2b* (*Wu et al., 2011*). Complete genotypes of figure panels are described in *Supplementary file 1*.

## Generation of *PlexB-Tag*

The *FRT-3xFLAG-6xStop-FRT-1xV5-6xStop* cassette (noted as Conditional Tag or *FRT-FLAG-Stop-FRT-V5-Stop* in *Figure 1H*) was synthesized as two gBlock fragments (Integrated DNA Technologies, Coralville, IA, USA) and assembled into the *pBS-KS-attB2-SA(0)-T2A-GAL4-Hsp70* vector (*Diao et al., 2015*) to replace the original phase spacer and the *T2A-GAL4* coding region by NEBuilder HiFi DNA assembly master mix (New England Biolabs, Ipswich, MA, USA). The *PlexB* coding sequence after the first exon was amplified from *PlexB* full-length cDNA (*Joo et al., 2013*) by Q5 hot-start high-fidelity DNA polymerase (New England Biolabs, Ipswich, MA, USA) and inserted into *pBS-KS-attB2-SA-Conditional Tag-Hsp70* in front of the Conditional Tag sequence. The three prime untranslated region (3'-UTR) of *PlexB* was amplified from the *w*[1118] genomic DNA, extracted by DNeasy blood and tissue kit (QIAGEN, Hilden, Germany), to replace the original *Hsp70* terminator in the vector *pBS-KS-attB2-SA-PlexB cDNA after Exon 1-Conditional Tag-Hsp70*. The final vector *pBS-KS-attB2-SA-PlexB cDNA after Exon 1-Conditional Tag-PlexB 3'-UTR* was transformed into NEB stable competent *E. coli* (New England Biolabs, Ipswich, MA, USA), extracted by QIAprep spin miniprep kit (QIAGEN, Hilden, Germany), and verified by full-length sequencing (Elim Biopharmaceuticals, Hayward, CA, USA). This final vector was co-injected with a *phiC31* plasmid into *PlexB*[MI15559] embryos. All *yellow*[−] progenies were individually balanced by P{ActGFP}unc-13[GJ]/In(4)ci[D], ci[D], pan[ciD] (#9549, Bloomington *Drosophila* Stock Center, Bloomington, IN, USA). The genomic sequence flanking the MiMIC cassette was sequenced to confirm the insertion and its orientation.

## Generation of *Or92a-rCD2*

The promoter sequence of the *Or92a* gene (*Fishilevich and Vosshall, 2005*) was amplified from the *w*[1118] genomic DNA by Phusion high-fidelity DNA polymerase (New England Biolabs, Ipswich, MA, USA) and inserted upstream of the rat CD2 coding sequence in a P-element vector (*Borkowski et al., 1995*). After verified by sequencing, this construct was co-injected with a *Δ2–3* helper plasmid into *w*[1118] embryos. All *white*[+] progenies were individually balanced. One inserted into the second chromosome was verified and used in this study.

## Immunocytochemistry

Dissection and immunostaining (except the staining of *PlexB-Tag*, see below for details) of fly brains and antennae were performed according to previously described methods (*Li et al., 2016*; *Wu et al., 2017*; *Wu and Luo, 2006*). Briefly, the brains/antennae were dissected in PBS (phosphate buffered saline; Thermo Fisher, Waltham, MA, USA) and then fixed in 4% paraformaldehyde (Electron Microscopy Sciences, Hatfield, PA, USA) in PBS with 0.015% Triton X-100 (Sigma-Aldrich, St. Louis, MO, USA) for 20 min on a nutator at room temperature. Fixed brains/antennae were washed with PBST (0.3% Triton X-100 in PBS) four times, each time nutating for 20 min. The brains/antennae were then blocked in 5% normal donkey serum (Jackson ImmunoResearch, West Grove, PA, USA) in PBST for 1 hr at room temperature or overnight at 4°C on a nutator. Primary antibodies were diluted in the blocking solution and incubated with brains/antennae for 36–48 hr on a 4°C nutator. After washed with PBST four times, each time nutating for 20 min, brains/antennae were incubated with secondary antibodies diluted in the blocking solution and nutated in the dark for 36–48 hr at 4°C.

Brains/antennae were then washed again with PBST four times, each time nutating for 20 min. Immunostained brains/antennae were mounted with SlowFade antifade reagent (Thermo Fisher, Waltham, MA, USA) and stored at 4°C before imaging.

For the staining of *PlexB-Tag*, the routine protocol described above failed to detect FLAG or V5 signal from the background, likely due to the low expression of endogenous PlexB proteins in vivo. Alexa 488 Tyramide SuperBoost kit (Thermo Fisher, Waltham, MA, USA) was used to amplify the immunostaining signal by following the manufacture's protocol. Briefly, the brains were dissected, fixed, and washed as described above. After rinsed with PBS twice, the brains were incubated with 3% hydrogen peroxide for 1 hr at room temperature, to quench the activity of endogenous peroxidases and then washed with PBS three times. After blocked in 10% goat serum for 1 hr at room temperature, the brains were nutated in primary antibodies diluted in 10% goat serum for 36–48 hr at 4°C. After washed with PBST four times, each time nutating for 20 min, the brains were incubated with the poly-HRP-conjugated secondary antibody provided in the kit and nutated for 36–48 hr at 4°C. Then, the brains were washed with PBST four times, each time nutating for 20 min, followed by two rounds of fast rinsing in PBS. The tyramide working solution and the quenching buffer were made freshly according to the kit's recipe. The brains were incubated with the tyramide solution for 5 min at room temperature and immediately washed with the quenching buffer three times, followed by four rounds of thorough washing with PBST. Stained brains were mounted with SlowFade antifade reagent (Thermo Fisher, Waltham, MA, USA) and stored at 4°C before imaging.

Primary antibodies used in this study include: rat anti-NCad (1:40; DN-Ex#8, Developmental Studies Hybridoma Bank, Iowa City, IA, USA), mouse anti-ELAV (1:40; 9F8A9, Developmental Studies Hybridoma Bank, Iowa City, IA, USA), chicken anti-GFP (1:1000; GFP-1020, Aves Labs, Tigard, OR, USA), rabbit anti-DsRed (1:200; 632496, Clontech, Mountain View, CA, USA), mouse anti-rat CD2 (1:200; OX-34, Bio-Rad, Hercules, CA, USA), mouse anti-FLAG (1:100; M2, Sigma-Aldrich, St. Louis, MO, USA), mouse anti-V5 (1:100; R960-25, Thermo Fisher, Waltham, MA, USA), and rabbit anti-Sema2b (1:500; *Sweeney et al., 2011*). Donkey secondary antibodies conjugated to Alexa Fluor 405/488/568/647 or FITC (Jackson ImmunoResearch, West Grove, PA, USA or Thermo Fisher, Waltham, MA, USA) were used at 1:250.

## Image acquisition, processing, and quantification

Images were acquired by a Zeiss LSM 780 laser-scanning confocal microscope (Carl Zeiss, Oberkochen, Germany), with a 40x/1.4 Plan-Apochromat oil objective (Carl Zeiss, Oberkochen, Germany). Confocal z-stacks were obtained by 1 μm intervals at the resolution of 512 × 512.

For quantification of ORN axon trajectories at 24hAPF, the z-stack of an antennal lobe was collapsed to one image by maximum intensity projection (ZEN software, Carl Zeiss, Oberkochen, Germany). Each antennal lobe was divided into two halves (DL and VM) by the line from the ORN axon entry point to the commissure merging point. The fluorescence intensities of the DL and VM halves and an area outside of the antennal lobe (background) were measured by ImageJ (NIH, Bethesda, MD, USA). Background fluorescence intensity was deducted to obtain the corrected intensities of the DL and VM axon trajectories. The DL/VM ratio was calculated by Excel (Microsoft, Redmond, WA, USA).

For quantification of the ectopic VM axons in *Figure 2F*, if DA1 axons stayed only at the DL edge of the antennal lobe (as in *Figure 2D*), the antennal lobe was counted as 'Only DL axons'. If DA1 axons show up inside of the antennal lobe (as in *Figure 2E*), it was counted as 'VM+DL axons'. For quantification of the glomerular targeting in *Figure 4* and *Figure 5*, the numbers of antennal lobes with corresponding phenotypes were counted. Data were presented as a percentage and no statistical test was performed.

Images were exported as maximum projections or single confocal sections by ZEN (Carl Zeiss, Oberkochen, Germany) in the format of TIFF. Photoshop (Adobe, San Jose, CA, USA) was used for image rotation and cropping. Illustrator (Adobe, San Jose, CA, USA) was used to make diagrams and assemble figures.

## Western blotting

24hAPF brains of *PlexB-Tag* flies were dissected in the Schneider's *Drosophila* medium (Thermo Fisher, Waltham, MA, USA) and snap frozen in liquid nitrogen before stored at –80°C. The sample

was lysed on ice in pre-cooled RIPA buffer (Thermo Fisher, Waltham, MA, USA) with protease inhibitors (100X Halt cocktail; Thermo Fisher, Waltham, MA, USA) and then rotated for 1 hr at 4°C. After centrifugation for 15 min at 16000 RCF at 4°C, the supernatant was collected and kept on ice. NuPAGE LDS sample buffer and NuPAGE reducing agent (Thermo Fisher, Waltham, MA, USA) were added to the sample, followed by heating at 70°C for 10 min. Precision Plus Protein Kaleidoscope prestained protein standard (Bio-Rad, Hercules, CA, USA) was used as the molecular weight marker. Electrophoresis with the NuPAGE Tris-acetate gel and transferring to the PVDF membrane (Thermo Fisher, Waltham, MA, USA) were performed according to the manufacture's protocols. The membrane was blocked by 5% bovine serum albumin (Sigma-Aldrich, St. Louis, MO, USA) in TBST (25 mM Tris, 0.15M NaCl, 0.05% Tween-20, pH 7.5; Thermo Fisher, Waltham, MA, USA) and incubated with the primary antibody (mouse anti-FLAG, 1:500, M2, Sigma-Aldrich, St. Louis, MO, USA) in 5% BSA-TBST overnight on a 4°C orbital shaker. After washing with TBST, the membrane was incubated with the secondary antibody (goat anti-mouse HRP-conjugated, 1:5000, ab97023, Abcam, Cambridge, UK) for 1 hr on an orbital shaker at room temperature. The signal was developed with Super-Signal West Pico PLUS chemiluminescent substrate (Thermo Fisher, Waltham, MA, USA) and captured by the ChemiDoc XRS+ system (Bio-Rad, Hercules, CA, USA).

## Statistical analysis

No statistical methods were used to determine sample sizes, but our sample sizes were similar to those generally employed in the field. Antennal lobes damaged in dissection were excluded from analysis; otherwise, all samples were included. Data collection and analysis were not performed blind to the conditions of the experiments. GraphPad Prism (GraphPad Software, La Jolla, CA, USA) was used for statistical analysis and plotting. Significance between two groups was determined by an unpaired, two-tailed $t$-test. Significance among multiple groups was determined by one-way ANOVA with Tukey's test for multiple comparisons. Significance of contingency tables was determined by Fisher's exact test.

## Acknowledgements

We thank Hugo Bellen, Yuh-Nung Jan, and Alex Kolodkin for the kind gifts of reagents, the Bloomington *Drosophila* Stock Center and the Vienna *Drosophila* Resource Center for fly lines, the Developmental Studies Hybridoma Bank for antibodies, and Addgene for plasmids. We acknowledge Dominic Berns, Thomas Clandinin, Xiaojing Gao, K Christopher Garcia, Yanyang Ge, Shuo Han, Wei-Hsiang Huang, William Joo, Alex Kolodkin, Jan Lui, Shan Meltzer, Timothy Mosca, Jing Ren, Kang Shen, Andrew Shuster, Michael Simon, Xin Wang, Alex Ward, Anthony Xie, Qijing Xie, Susan Younger, Wei Zhang, and all Luo lab members for technical support, insightful advice, and/or critical comments on this study. We thank Alex Kolodkin for his support of Xiaojun Xie. We also acknowledge Stephanie Wheaton for administrative assistance.

## Additional information

### Funding

| Funder | Grant reference number | Author |
|---|---|---|
| National Institutes of Health | R01-DC005982 | Liqun Luo |
| Howard Hughes Medical Institute | | Liqun Luo |
| Stanford University | Vanessa Kong Kerzner Graduate Fellowship | Jiefu Li |
| Genentech Foundation | Genentech Foundation Predoctoral Fellowship | Jiefu Li |
| Stanford University | Stanford Neuroscience Institute Interdisciplinary Scholar | Hongjie Li |

The funders had no role in study design, data collection and interpretation, or the decision to submit the work for publication.

## Author contributions
Jiefu Li, Conceptualization, Resources, Data curation, Formal analysis, Validation, Investigation, Visualization, Methodology, Writing—original draft, Writing—review and editing; Ricardo Guajardo, Resources, Data curation, Formal analysis, Validation, Investigation; Chuanyun Xu, Bing Wu, Hongjie Li, Tongchao Li, David J Luginbuhl, Resources, Investigation; Xiaojun Xie, Resources; Liqun Luo, Conceptualization, Resources, Data curation, Formal analysis, Supervision, Funding acquisition, Validation, Visualization, Methodology, Writing—original draft, Project administration, Writing—review and editing

## Author ORCIDs
Jiefu Li http://orcid.org/0000-0002-0062-4652
Xiaojun Xie http://orcid.org/0000-0003-3459-6095
Liqun Luo http://orcid.org/0000-0001-5467-9264

## Decision letter and Author response
Decision letter https://doi.org/10.7554/eLife.39088.015
Author response https://doi.org/10.7554/eLife.39088.016

# Additional files

## Supplementary files
• Supplementary file 1. Genotypes of flies in each experiment.
DOI: https://doi.org/10.7554/eLife.39088.012

• Transparent reporting form
DOI: https://doi.org/10.7554/eLife.39088.013

## Data availability
All data generated or analysed during this study are included in the manuscript and supporting files.

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
