## [Decision Letter]

Thank you for submitting your article "Stepwise wiring of the *Drosophila* olfactory map requires optimal Plexin B levels" for consideration by *eLife*. Your article has been reviewed by three peer reviewers, one of whom is a member of our Board of Reviewing Editors, and the evaluation has been overseen by including Kristin Scott as the Reviewing Editor and K VijayRaghavan as the Senior Editor. The following individuals involved in review of your submission have agreed to reveal their identity: Bing Ye (Reviewer #2) and Takahiro Chihara (Reviewer #3).

The reviewers have discussed the reviews with one another and the Reviewing Editor has drafted this decision to help you prepare a revised submission.

Summary:

This manuscript by Li et al. demonstrates the dual role of PlexB in trajectory choice and target selection, two consecutive steps for ORNs to connect to their proper postsynaptic targets in *Drosophila* olfactory system. The authors generated fly lines that allowed conditional tagging of endogenous PlexB, and used them to examine PlexB expression and distribution specifically in ORNs. This critical approach led to the discovery that ORN PlexB protein distribution was uneven in the antennal lobe and changed when axon development switched from trajectory choice to target selection. The authors further elegantly applied *Drosophila* genetics to demonstrate that the levels of PlexB regulated both steps, independently. The experiments are carefully designed and executed, and the presentation is clear. The study is important because it offers an understanding of how complicated neural networks are established with a limited number of cues.

Essential revisions:

1) The notion that optimal levels of PlexB are required is not fully demonstrated in this study. This seems like one possible model based on the lof and OE studies but the actual relationship between levels and targeting needs further exploration (by quantifying levels in knockdowns and testing additional concentrations) or the conclusions that optimal levels are required should be toned down.

2) The PlexB overexpression phenotypically resembled loss of PlexB, shifting the ORN trajectory to the DL side. In wild type, are there any ORNs that do not express PlexB? Are such ORNs in the DL trajectory and targeting laterally-located glomeruli?

3) Related to Figure 2E: Loss of PlexB causes the shift of ORN bundle to the DL side. What is the phenotype of the *Or67d-QF*ORNs in the PlexB mutant?

4) The bulk analyses of axons, while informative, may hide single axon targeting defects. For example, is it possible that there are more axons at 24hrs that stop short in the PlexB knockout or travel further in the OE? Single cell mosaics would be informative, but if beyond the scope of the study, then discussion of caveats with bulk analyses is warranted.

5) The post-transcriptional and post-translational regulation of PlexB is very intriguing. If the authors have any insights into PlexB regulation, this would be interesting to discuss.

---

## [Author Response]

Essential revisions:1) The notion that optimal levels of PlexB are required is not fully demonstrated in this study. This seems like one possible model based on the lof and OE studies but the actual relationship between levels and targeting needs further exploration (by quantifying levels in knockdowns and testing additional concentrations) or the conclusions that optimal levels are required should be toned down.

We agree with the reviewer that we have not quantitatively revealed where the “optimal” point of PlexB levels is. Our observation that moderate PlexB knockdown causes more DL-to-VM axon shift than severe knockdown (Figure 2F) indicates the existence of an “optimal” PlexB level for the VM trajectory choice, although the exact “optimal” point remains unknown. Quantification and comparison of ORN PlexB protein concentrations under different knockdown conditions is technically challenging. To be more conservative, we have removed the claims of “optimal PlexB levels” and modified the conclusions.

2) The PlexB overexpression phenotypically resembled loss of PlexB, shifting the ORN trajectory to the DL side. In wild type, are there any ORNs that do not express PlexB? Are such ORNs in the DL trajectory and targeting laterally-located glomeruli?

To test if any ORN does not express PlexB, we crossed *PlexB-GAL4* with a nuclear DsRed reporter line and examined the third segments of antennae, where individual ORN somas can be unambiguously identified by ELAV immunostaining. We found that *PlexB-GAL4* labels most ORNs (92.3% at 24hAPF as in Figure 1—figure supplement 1D; 98.7% at 48hAPF as in Figure 3—figure supplement 1A). We have included the data in the revised manuscript (subsection “Conditional tagging reveals that ORN axons of the DL trajectory possess a higher level of PlexB than the VM axons”, second paragraph; subsection “Differential distribution of PlexB proteins in ORN axons at the stage of glomerular selection”, second paragraph).

3) Related to Figure 2E: Loss of PlexB causes the shift of ORN bundle to the DL side. What is the phenotype of the Or67d-QF ORNs in the PlexB mutant?

We previously reported that PlexB mutant does not change the DL trajectory choice and DA1 glomerular targeting of Or67d+ ORNs (Joo et al., 2013). It is consistent with our observation that severe PlexB knockdown leads to more DL targeting of DA1 axons (Figure 2F). We have integrated this into the manuscript (subsection “Moderate levels of PlexB knockdown cause DL ORN axons to shift to the VM trajectory”, last paragraph).

4) The bulk analyses of axons, while informative, may hide single axon targeting defects. For example, is it possible that there are more axons at 24hrs that stop short in the PlexB knockout or travel further in the OE? Single cell mosaics would be informative, but if beyond the scope of the study, then discussion of caveats with bulk analyses is warranted.

We agree with the reviewer that single-cell mosaic analysis is complementary to bulk genetic manipulation and can better reveal the cell-autonomous function of the gene of interest. However, *PlexB* is localized on the fourth chromosome, precluding mosaic analysis on it. We have added discussion on this issue (subsection “Level-dependent effects of PlexB”, second paragraph). Regarding to the question above, no developmental delay or acceleration of ORN axons was observed in PlexB mutant or OE, at least at the bulk level.

5) The post-transcriptional and post-translational regulation of PlexB is very intriguing. If the authors have any insights into PlexB regulation, this would be interesting to discuss.

We agree with the reviewer that this is an important question. With the observation that Sema2b antagonizes the effect of high-level PlexB, we hypothesized that Sema2b may regulate PlexB level and thus achieves its antagonistic effect. However, our biochemical assays with *PlexB-Tag* pupal brains did not provide supporting evidence. So unfortunately, we do not know how the PlexB level is regulated. We added a sentence in Discussion to state that the mechanism requires future investigation (subsection “PlexB expression and protein distribution in developing olfactory circuits”, last paragraph).

In addition to changes made in response to reviewers’ comments, we added *PlexB-Tag* staining at 30hAPF and 36hAPF (Figure 3—figure supplement 1B, C) to better reveal the PlexB distribution transitioning from 24hAPF to 48hAPF. We also performed Fisher’s exact test on the data of Figure 2F and added the results.